# Understanding the Influence of a Bifunctional Polyethylene Glycol Derivative in Protein Corona Formation around Iron Oxide Nanoparticles

**DOI:** 10.3390/ma12142218

**Published:** 2019-07-10

**Authors:** Amalia Ruiz, Adán Alpízar, Lilianne Beola, Carmen Rubio, Helena Gavilán, Marzia Marciello, Ildefonso Rodríguez-Ramiro, Sergio Ciordia, Christopher J. Morris, María del Puerto Morales

**Affiliations:** 1School of Pharmacy, University of East Anglia, Norwich Research Park, Norwich NR4 7TJ, UK; 2Centro Nacional de Biotecnología (CNB)/CSIC, Darwin, 3, 28049 Madrid, Spain; 3Instituto de Ciencia de Materiales de Aragón (ICMA), CSIC/Universidad de Zaragoza, C/Pedro Cerbuna 12, 50009 Zaragoza, Spain; 4Centro de Biología Molecular “Severo Ochoa” (CBMSO)/UAM-CSIC, Nicolás Cabrera, 1, 28049 Madrid, Spain; 5Instituto de Ciencia de Materiales de Madrid (ICMM)/CSIC, Sor Juana Inés de la Cruz 3, Cantoblanco, 28049 Madrid, Spain; 6Faculty of Pharmacy, Complutense University of Madrid (UCM), Plaza Ramón y Cajal s/n, 28040 Madrid, Spain; 7School of Medicine, University of East Anglia, Norwich Research Park, Norwich NR4 7TJ, UK

**Keywords:** iron oxide nanoparticles, protein corona, PEG-coated nanoparticles

## Abstract

Superparamagnetic iron oxide nanoparticles are one of the most prominent agents used in theranostic applications, with MRI imaging the main application assessed. The biomolecular interface formed on the surface of a nanoparticle in a biological medium determines its behaviour in vitro and in vivo. In this study, we have compared the formation of the protein corona on highly monodisperse iron oxide nanoparticles with two different coatings, dimercaptosuccinic acid (DMSA), and after conjugation, with a bifunctional polyethylene glycol (PEG)-derived molecule (2000 Da) in the presence of Wistar rat plasma. The protein fingerprints around the nanoparticles were analysed in an extensive proteomic study. The results presented in this work indicate that the composition of the protein corona is very difficult to predict. Proteins from different functional categories—cell components, lipoproteins, complement, coagulation, immunoglobulins, enzymes and transport proteins—were identified in all samples with very small variability. Although both types of nanoparticles have similar amounts of bonded proteins, very slight differences in the composition of the corona might explain the variation observed in the uptake and biotransformation of these nanoparticles in Caco-2 and RAW 264.7 cells. Cytotoxicity was also studied using a standard 3-(4,5-dimethylthiazol-2-yl)-2,5-diphenyl tetrazolium bromide assay. Controlling nanoparticles’ reactivity to the biological environment by deciding on its surface functionalization may suggest new routes in the control of the biodistribution, biodegradation and clearance of multifunctional nanomedicines.

## 1. Introduction

Iron oxide nanoparticles (NPs) are considered promising tools for theragnostic applications in biomedicine. Fourteen clinical protocols have been registered on clinicaltrials.gov, with magnetic resonance imaging (MRI) the main application [1]. Other studies in different clinically relevant applications, ranging from drug delivery, magnetic hyperthermia, tissue repair to in vitro diagnostics indicate their biomedical potential but also the need for improvement prior to their approval by regulatory agencies [2]. Issues related to blood circulation time, bioavailability, biocompatibility, toxicological and immunological response are directly linked to the physico–chemical characteristics and colloidal properties of the NPs [3]. For example, particles with sizes below 10 nm are rapidly cleared by renal excretion. Meanwhile, particles with sizes higher than 200 nm are easily phagocytized by macrophages in the reticuloendothelial system, leading to a decrease in blood circulation time. A large variety of NPs’ surface coatings, using different polymers, have been developed to improve their biocompatibility and colloidal stability in biological environments.

Years ago, the practice of PEGylation, or covalent attachment of polyethylene glycol (PEG) to proteins, peptides, micelles, liposomes and nanoparticles was adopted by many researchers with the aim of improving their bioavailability, stability, safety and reducing immunogenicity. Polyethylene glycol is a safe and low-cost additive for pharmaceutical and cosmetic industries that has been approved by regulatory agencies for human intravenous, oral and dermal applications [4]. In fact, PEGylation is one of the most favoured ways to modify the surface of nanoparticles for biomedical applications including intracellular targeting, antitumoral delivery systems and diagnostics [5,6,7,8,9]. It has been shown that PEGylation of NPs improves their biocompatibility and reduces uptake by the reticuloendothelial system (RES) [10]. Polyethylene glycol solvation properties also have an effect on the efficiency of NPs used as MRI contrast agents [11,12]. Thus, MRI relaxation times, T_1_ and T_2_ signals, are influenced by, among other factors, the concentration of water molecules in the proximity of the magnetic nanoparticle, and therefore, on the solvation abilities of the coating. Conjugation to bifunctional PEG derivatives like the O,O’-bis(2-aminoethyl) polyethylene glycol can also offer grafting sites for covalent immobilization of targeting moieties such as proteins, peptides or antibodies [13]. 

However, it is well known that when NPs are exposed to a biological medium, biomolecule adsorption occurs immediately, altering their physico–chemical characteristics and giving rise to cellular recognition of the nanoparticles by phagocytes or inflammatory cells (opsonization), which significantly reduces the efficiency of active targeting and may cause problems in their application in vivo [14]. After intravenous administration, the proteins present in the blood interact with the NPs’ surface, forming the so-called protein corona [15]. Numerous studies have shown that the protein corona defines the biological identity of nanoparticles which mandates a new characterization strategy that is not only related to their physico–chemical properties from the bare material but also takes into account the new derived properties from the biological environment [16]. 

Significant progress has been made towards understanding protein corona formation and its evolution around NPs. However, human serum is a complex fluid that contains about 3700 different proteins with concentrations up to 70 mg/mL [17]. Among them, the most abundant ones are albumin, immunoglobulin G (IgG) and α2-macroglobulin that represent 97% of the total protein content, whereas the remaining 3% is a mixture composed of other proteins [18]. When NPs come into contact with a biological fluid, these proteins are adsorbed on to their surface, forming a stable, hard core composed of biological macromolecules (“hard corona”) that interact strongly with the surface and an outer layer that is formed of loosely bound proteins (“soft corona”) that are in continuous exchange with the environment [19]. 

Extensive works have been published characterizing protein corona of several NPs based on different material, such as gold [20,21,22], silver [23,24], polystyrene [25,26,27,28,29], silica [25,28,29,30] and metal oxides like TiO_2_, ZnO, CoO and CeO [23,31]. Iron oxide nanoparticles are the only magnetic material approved for biomedical applications as negative contrast agents because they decrease the MRI signal intensity of the regions to which they are delivered [32]. Formulations for clinical use include both intravenous and oral administration. However, only a few studies about the formation and composition of the corona around these nanoparticles are available [33,34,35]. Polyvinyl alcohol, tetraethyl orthosilicate (TEOS), TiO_2_ and gold-coated iron oxide NPs have been used to study the influence of the surface charge and shell material on the protein adsorption [35]. The protein corona on PEGylated NPs has been studied in an effort to elucidate its influence on the reactivity and degradation of the NPs in vitro and in vivo [36,37]. However, contradictory results are obtained and many differences in the experimental conditions and the nature of the tested particles make it difficult to draw general conclusions. Another issue to highlight would be the effect of the qualitative and quantitative variations between species expressed as different isoelectric points or different molecular weights for the proteins [38] and their effect on the interaction with the nanoparticles’ surface. Further research is still required to understand NPs’ biodegradation associated with their protein corona and suggest new directions in the control of the degradation, biodistribution and clearance of multifunctional nanomedicines.

In previous studies, we developed an approach for the synthesis of magnetic nanoparticles coated with PEG-derived molecules, which avoided aggregation while maintaining high-quality magnetic and relaxometric properties such as MRI contrast agents [39,40]. In this work, we have investigated, through extensive proteomic analysis, the formation and composition of the protein corona around magnetic nanoparticles with two different coatings: one short and highly charged small molecule, such as DMSA, and the surface modification of a diamine (PEG)-derived molecule (2000 Da) widely used for providing a long circulation time [40,41].

## 2. Materials and Methods 

### 2.1. Nanoparticle Synthesis

Oleic acid-coated nanoparticles were synthesized by thermal decomposition in a procedure previously described [42]. To render hydrophilic nanoparticles, a ligand exchange reaction was carried out. The surfaces of the nanoparticles were modified with dimercaptosuccinic acid (DMSA) and oleic acid was removed (NP-DMSA) by a method published elsewhere [43,44].

Diamine-functionalized PEG was covalently coupled to NP-DMSA via ethyl-3-(3-dimethylaminopropyl)-carbodiimide (EDC)-mediated reaction using O,O´-bis(2-aminoethyl) polyethylene glycol, 2000 Da (PEG-(NH_2_)_2(2000)_). The conjugation reaction has previously been published [39].

### 2.2. Nanoparticle Characterization

Particle size and shape were studied using a 200-KeV JEOL-2000 FXII microscope (Tokyo, Japan). The size distributions were determined by manual measurement of more than 100 particles using the public domain software ImageJ6. Colloidal properties were characterized by dynamic light scattering (DLS) using a Nanosizer ZS (Malvern Instruments, Malvern, UK). The Z-average values in intensity were used as mean hydrodynamic size. The ζ-potential was measured using a solution of 20 mM of KNO_3_. Both HNO_3_ or KOH were added to adjust the pH. Fourier transform infrared spectroscopy (FTIR) spectra were acquired using a Nicolet 20 SXC FTIR (Thermo Fisher Scientific Waltham, MA, USA). Infrared spectra of the NPs were recorded between 250 and 4000 cm^−1^. Thermogravimetric analysis was carried out in a Seiko TG/ATD 320 U, SSC 5200 (Tokyo, Japan). The analysis was performed from room temperature to 1100 °C, at 10 °C /min with an air flow rate of 100 mL/min.

### 2.3. Nanoparticle Incubation with Plasma

Blood was taken from 10 different male Wistar rats (from 12–16 weeks old) weighing 350–400 g. Blood samples (10 tubes of 3 mL) were centrifuged for 5 min at 800 RCF to pellet the red and white blood cells. The plasma supernatant was pooled and stored at −80 °C until used.

Iron oxide nanoparticles (300 µg) were diluted in phosphate-buffered saline (PBS) solution to a final concentration of 1 mg/mL. Pooled plasma (500 µL) was then added to each tube and samples were incubated on a rotary mixer for 10, 20 and 30 min at 37 °C. Following incubation, nanoparticles were separated magnetically and washed 3 times with 2 mL of sterile PBS and the final samples were processed for analysis. The evolution of colloidal properties was characterized by dynamic light scattering (DLS) using a Nanosizer ZS (Malvern).

### 2.4. Polyacrylamide Gel Electrophoresis

Preliminary analysis of the soft and hard corona around the particles was carried out by electrophoresis SDS-PAGE. Particles (300 µg) incubated at 10, 20 and 30 min with bound proteins were washed sequentially with 2 mL of different buffers: (a) 50 mM HEPES 0.1% N-octyl-β-D-glucoside (OGP) pH 7, (b) 100 mM NaAc 0.1% OGP pH 5. Each sample was shaken in a laboratory tube rotator with 2 mL of the corresponding buffer solution for 30 min and was centrifuged 10 min at 4 °C in a microcentrifuge. Then, Laemmli buffer was added to the mixture and the samples were loaded on a Mini PROTEAN TGX^TM^ 10% pre-cast gel (BioRad, Hercules, CA, USA). The electrophoresis was carried out under reducing conditions. The gels were stained with Coomassie Brilliant Blue R-250 (Thermo Fisher Scientific, Waltham, MA, USA). Gel images were processed by ImageJ6.

### 2.5. Liquid Chromatography and Mass Spectrometry Analysis

Particles (300 µg) with bound proteins were washed sequentially for 30 min with different buffers: (a) 50 mM HEPES 0.1% OGP pH 7, (b) 100 mM NaAc 0.1% OGP pH 5. Each sample was shaken in a laboratory tube rotator with 2 mL of the corresponding buffer solution for 30 min and was centrifuged 10 min at 4 °C in a microcentrifuge. The supernatant was collected and stored at −80 °C until analysis.

The eluted proteins in each step of the purification process were precipitated by a methanol/chloroform protocol, [45] quantified by Pierce 660 nm reagent (Thermo Scientific) in 8 M urea, 25 mM ammonium bicarbonate ((NH_4_)HCO_3_), reduced with 50 mM of Tris(2-carboxyethyl) phosphine (TCEP), alkylated with 20 mM of methyl methanethiosulfonate (MMTS) and finally digested with trypsin (1:20 enzyme:protein, weight ratio) according to Reference [46]. The digest was passed through a SEP-PAK C18 column prior to analysis by mass spectrometry (Waters, Milford, MA, USA).

The samples digested were subjected to electrospray ionization mass spectrometry, 1D-nano LC ESI-MSMS analysis using a nano-liquid chromatography system (Eksigent Technologies nanoLC Ultra 1D plus, AB SCIEX, Foster City, CA, USA) coupled to a high-speed Triple TOF 5600 mass spectrometer (AB SCIEX, Foster City, CA, USA) with a Nanospray III source. Full description of the methodology has been published elsewhere [47]. 

### 2.6. Data Analysis: MS/MS Ion Search and Peptide Identification

Raw data was processed with PeakView software (version 1.1, AB Sciex) to generate mascot generic format (MGF) files that were used as input to run the searches against the Uniprot *Rattus norvegicus* database, containing 7932 protein coding genes entries and using the Mascot search engine v.2.5 (Matrix Science, London, UK). Search parameters were set as follows: enzyme, trypsin; allowed missed cleavages, 2 fixed modifications, beta-methylthiolation of cysteine; variable modifications, oxidation of methionine. Peptide mass tolerance was set to ±25 ppm for precursors and 0.05 Da for fragment masses. The confidence interval for protein identification was set to ≥95% (*p* < 0.05) and only peptides with an individual ion score or expected cut-off above 20 were considered correctly identified. Only proteins having at least two identified peptides were considered as significant.

### 2.7. Cell Culture

Caco-2 and RAW 264.7 were obtained from American Type Culture Collection (Manassas, VA, USA). Cells were cultured in Dulbecco’s Modified Eagle Medium (DMEM) supplemented with 10% fetal bovine serum and 2% penicillin-streptomycin in a humidified incubator (37 °C, 5% CO_2_). 

### 2.8. Cytotoxicity Assay (MTT)

For toxicity experiments, cells were seeded in 96 well plates (approximately 10,000 cells/well, 0.2 mL/well). Cell viability was determined using the standard 3-(4,5-dimethylthiazol-2-yl)-2,5-diphenyl tetrazolium bromide (MTT) assay 24 h after exposure to NPs in a procedure described elsewhere [48]. 

### 2.9. Iron Uptake

For iron uptake experiments, Caco-2 cells were seeded onto collagen-coated 12 well plates (Bio-Greiner, UK) at a density of 10,000 cells per well suspended in 1 mL of supplemented DMEM. Detailed description of the protocol has previously been published [48,49]. The day of the experiment, the nanoparticles were diluted in low-iron MEM to obtain a 250 µM final iron concentration and subsequently Caco-2 cells were exposed for 24 h with the treatments.

### 2.10. Nitric Oxide Production

To determine NO production in the presence of the nanoparticles, RAW 264.7 macrophages were seeded onto 96 well plates (10,000 cells/well) and grown overnight. Culture medium was removed and replaced with 0.1 mL fresh medium supplemented with 500 ng *E. coli* lipopolysaccharide (LPS) alone (positive control) or NP-DMSA and NP-PEG-(NH_2_)_2(2000)_ at 50 or 250 µM. After 24 h, media was removed, centrifuged briefly (200× *g*, 5 min) to remove any cellular material and then assayed for nitrite concentration as an indicator of nitric oxide production by iNOS. Briefly, 0.1 mL supernatant was mixed with 0.05 mL sulfanilamide (1% w/v in 1% v/v HCl) for 15 min and then 0.05 mL of *N*-1-napthylethylenediamine dihydrochloride (0.1 % w/v in 1% HCl). After 15 min incubation at room temperature, aliquots were transferred to a fresh 96 well plate and A_550nm_ measured using a plate reader. Sample nitrite concentrations were determined from a sodium nitrite calibration curve.

## 3. Results

### 3.1. Synthesis and Characterization of DMSA and PEG-Coated Nanoparticles

Uniform magnetite NPs with a core size of 10 nm and polydispersity index (PDI) of 0.15 were synthesized by thermal decomposition of the Fe(acac)_3_ precursor in 1-octadecene. The NPs showed a round morphology and were well dispersed due to the presence of oleic acid around the particles (Figure 1a). However, these particles are hydrophobic. To render hydrophilic nanoparticles suitable for biological applications, oleic acid was substituted with DMSA via a ligand exchange reaction (NP-DMSA). After coating with DMSA, the hydrodynamic size was higher than sizes measured by TEM, indicating some degree of agglomeration after the surface modification. Hydrodynamic size for NP-DMSA was 32 nm, with a polydispersity degree (PDI) lower than 0.25. The ζ-potential of NP-DMSA at pH 7.4 was negative (−35 mV) due to the presence of carboxylic groups of DMSA on the surface of the particles (Figure 1b).

To study the influence of a bifunctional polyethylene glycol derivative in the protein corona formation around the NPs, we conjugated covalently a diamine PEG derivative (2000 Da) to the carboxylic groups of DMSA on the Nps surface via EDC-mediated coupling reaction [39]. The conjugation efficiency for the diamine-PEG derivative was confirmed by FTIR (Figure 1c).

For the NP-DMSA sample, the spectrum was characterized by intense peaks assigned to Fe–O vibration in the region of 550–600 cm^−1^ [50] and a broad peak between 3000–3500 cm^−1^ corresponding to surface hydroxyl groups [51,52]. After PEG grafting, some bands appeared at 1354 and 1102 cm^−1^, indicating asymmetric and symmetric stretching of C–O–C [53,54]. For nanoparticles conjugated to the bifunctional PEG derivative, an intense band appeared between 1000–1200 cm^−1^ after surface modification, indicating the presence of aliphatic polyether (C–O–C) and amide carbonyl vibration at 1640 and 1556 cm^−1^ [55,56]. This indicates covalent bonding of PEG-(NH_2_)_2(2000)_ onto the DMSA-coated NPs. The 2800–2950 cm^−1^ peak, due to the –CH_2_– groups in the polymer conjugated sample, confirmed this [52]. After PEG conjugation, the average hydrodynamic size at physiological pH (pH 7.4) increased from 32 nm to 67 nm with a PDI of 0.18. Net surface charge decreased after PEG conjugation from approximately −35 to −15 mV (Figure 1b).

### 3.2. Monitoring Protein Corona Formation

Superparamagnetic iron oxide nanoparticles clinically used as MRI contrast agents are administered in two main routes. Gastromark is an oral suspension for bowel imaging while Feridex and Resovist, or the smaller Combidex, are intravenous formulations for liver/spleen or lymph node imaging, respectively. Initial interaction of plasma proteins with the NPs can dramatically change their size and agglomeration degree, dominating in an uncontrolled way their behavior in vivo [57]. The blood half-life of Ferumoxides (Feridex) is around 6 min before accumulating in the liver [58,59]. The pharmacokinetics of the NP-DMSA and NP-PEG-(NH_2_)_2(2000)_ after intravenous injection in Wistar rats was studied by measuring the decrease of 1/T_2_ (R_2_) signal in the blood. The R_2_ relaxation rate values decreased to basal levels in 10 min for NP-DMSA and in about 1 hour for NP-PEG-(NH_2_)_2(2000)_. Similar results were obtained in New Zealand white rabbits for NP-DMSA, while R_2_ decreased to basal values in 20 min for NP-PEG-(NH_2_)_2(2000)_ [39]. For these reasons, we decided to monitor the evolution of particles’ hydrodynamic size in the presence of plasma up to 30 min (Figure 2a).

A slight increase in the hydrodynamic diameter of the particles was observed. The major variation was showed for NP-DMSA (from 32 to 67 nm), probably due to its stronger negative surface charge compared with PEG-coated NPs. In the case of PEG-conjugated particles, they exhibited a size increase from 67 to 113 nm. As observed in Figure 2b, the size distribution of the nanoparticles remained relatively stable up to 30 min and below 200 nm in physiological conditions, which is an important requirement for in vivo intravenous administration. Kinetic studies showed a rapid formation of the protein corona. In all cases, the increase in hydrodynamic size was observed after 1 min of incubation of the NPs in presence of plasma. An atomistic description of such a complex mechanism of formation and evolution of the protein corona around the NPs is, at the moment, unfeasible. Nevertheless, some computational simulations and in vitro experiments have been used to describe the interaction between NPs and a few components of the plasma [60,61]. The mechanism of formation of the corona around the NP can be divided in to two main steps: (a) the NPs enter in the biological media and encounter biomolecules that first adsorb forming the initial corona and (b) the corona composition changes dynamically due to the competition between proteins. Also, protein–protein interactions may affect the overall kinetics of this process. For example, in the case of silica-coated NPs, the time needed to achieve 50% of the surface coated with Human serum albumin (HSA) is around 0.1 min, whilst if the analysis is done with HSA and transferrin, the time needed is approximately 0.2 min [61].

### 3.3. Protein Corona Characterization by Electrophoresis SDS-PAGE

It is generally accepted that the structure and composition of the corona changes over time [14,19]. The most abundant proteins bind first, but they are displaced by those with higher affinity and the resulting “hard corona” contains proteins in a relatively immobile layer. In the “soft corona”, a continuous exchange of molecules from the biological medium and the particle surface takes place, and this leads to a fast and persistent variation of the structure of the soft corona. This variability in composition of the soft corona makes a detailed investigation of the influence of the unstable part of the corona difficult, and thus, most work has been conducted on the investigation of the hard corona [62]. To clarify this issue, the principal objective of this experiment was the investigation of the influence of incubation time on the composition of the corona around the NPs.

Preliminary analysis of the soft and hard corona around the particles was carried out by SDS-PAGE. Equivalent amounts of NPs incubated with serum for 10, 20 and 30 min at 37 °C were eluted sequentially with buffers at different pH. We defined “soft corona” as the fraction of proteins eluted at pH 7 with a very small variation of ionic strength (50 mM HEPES 0.1% OGP, pH 7). The “hard corona” corresponds to the smaller fraction of proteins eluted with a more significant variation of pH and ionic strength (100 mM NaAc 0.1% OGP pH 5) (Appendix A).

Figure 3 shows the densitometric analysis of the Coomassie blue-stained PAGE bands obtained from hard and soft protein coronae. An increase in the protein content of each size fraction over time was visible in the soft corona, whereas in the hard corona, a more or less steady distribution was observed with a reduced number of proteins. We detected in the soft corona the greatest amount of protein bound to the different surfaces. Proteins detected at 10 min were also detected at the 30 min time point (Figure 3a,b). This result suggests that there was no displacement of previously adsorbed proteins over time [22]. The extrapolation of these results to an in vivo biological model could confirm the transient nature of the protein soft corona and could explain why it evolves when the biological environment changes [63].

### 3.4. Protein Corona Characterization by Mass Spectrometry

To determine the identity of the proteins involved in the corona formation on NPs incubated for 30 min, sequentially eluted proteins were digested and the peptides analysed by mass spectrometry. Venn diagrams were used to group the proteins identified, which were specific for each type of NPs and those that were common to both of them (Figure 4a,b). A total of 136 proteins were extracted and identified in both coronal layers of the NP-PEG-(NH_2_)_2(2000)_ versus 121 proteins for NP-DMSA. Complete proteomic data are listed in Appendix A. Sequence coverage refers to the degree to which the whole protein amino acid sequence has been covered by the peptides detected by mass spectrometry. In our case, we can confirm the presence of the protein after detection of two peptides with good score. Appendix A shows all the proteins identified by mass spectrometry, although proteins detected with only one peptide have to be considered at the limits of reliability.

Approximately 70%, for NP-DMSA, and 65%, for NP-PEG-(NH_2_)_2(2000)_, of the proteins detected in the soft corona were also detected in the hard corona of the nanoparticles. We also identified a higher number of specific proteins in the soft and hard corona of NP-PEG-(NH_2_)_2(2000)_ (Figure 4c,d), in contrast with different reports where PEGylation reduced protein adsorption to the surface of nanoparticles [22,64]. It is possible that the coating of PEG-modified nanoparticles where COOH and NH_2_ groups are available, facilitates the interaction with a larger number of proteins compared with NP-DMSA that only have COOH groups. Other factors to take into account could be the molecular weight of the PEG reagent used, the density of the polymer in the surface of the nanoparticles and the conformation adopted by the polymer [22,65]. In the case of Dobrovolskaia et al. [22], only when they used a 20 kDa PEG were significant differences in total bound protein between the un-PEGylated and PEG-coated nanoparticles observed. Similar results reducing the amount of proteins in the surface of iron oxide nanoparticles were observed by Stepien et al., when they conjugated a 5 kDa PEG to the surface of the nanoparticles [37].

It is important to mention that the protein fingerprints around the nanoparticles might vary to a certain degree when using different separation techniques [30]. For example, in our work, we detected the presence of serum albumin in the composition of soft and hard corona around the particles. This result is in agreement with Monopoli’s [29] work, where serum albumin was identified on the particles after washing three times with PBS. But it is in contrast with the report of Sakulkhu et al. [35], where serum albumin was eluted only after washing with high ionic strength solutions. There are different procedures for separating nanoparticle–protein complexes, including microfiltration, centrifugation, equilibrium dialysis and size-exclusion chromatography. Each method has advantages but also limitations in terms of sensitivity, resolution and reproducibility [30]. We selected magnetic separation to preserve the soft corona (proteins bound by weak interactions) around the nanoparticles by avoiding high shear forces. After this first separation process, washing steps with the buffers were made by shaking the samples and centrifugation. We selected this second approach to separate the protein corona from the surface of the nanoparticles because this is a stronger method of separation and it can also be used easily and is highly reproducible.

Derived from Appendix A shows the proteins detected in the soft and hard corona grouped according to their biological function. Proteins from different functional categories and/or cell components, lipoproteins, complement, coagulation, immunoglobulins, enzymes and transport proteins were identified in all samples. In order to investigate how the coating of the particles determines their biointeractions and how it governs their biological fate, the protein amounts were semi-quantitatively analysed by applying the spectral-counting method. The NpSpCk represents the total number of the MS spectra for all peptides attributed to a matched protein divided by its molecular weight (Mw) in Da. This ratio takes into account the protein size and evaluates the real contribution of each protein to the corona composition [29].

As shown in Figure 5, the relative abundance of proteins on the surface of both types of nanoparticles was comparable. In general, it is assumed that PEGylation extends blood circulation times through a significant reduction in protein binding, which is responsible for its “stealth” behaviour in vivo [66,67]. In line with this idea, different studies have correlated the amount of bound proteins (as an indicator of opsonization) with cell internalization or half-life circulation and measured reduced protein adsorption to nanoparticles incorporating PEGylated components [68,69,70]. In contrast, other studies have demonstrated that PEGylation increases half-life in the blood but does not reduce protein adsorption. In fact, several studies have reported that PEGylation increases protein binding [67].

Overall, PEGylation of the nanoparticles did not show a significant variation in their interactions with the proteins according to their biological function. However, the analysis of the relative abundance of the top three most represented proteins in the corona of the nanoparticles (serum albumin, apolipoprotein A-I and Ig kappa chain C region) showed that PEGylation reduced the content of the apolipoprotein and has an enriched amount of albumin in the surface compared with NP-DMSA (Figure 6). Proteins like immunoglobulins and apolipoproteins are considered opsonins that promote clearance of NPs via interaction with macrophage receptors [71]. In contrast, the enrichment in dysopsonins proteins like albumin promote prolonged circulation time in blood by blocking recognition by macrophages and thereby reducing their clearance [57].

To verify if this small variation in the protein corona of NP-DMSA and NP-PEG-(NH_2_)_2(2000)_ is correlated with their biological behaviour, a set of in vitro experiments were carried out. In vitro cytotoxicity of the nanoparticles was evaluated by means of the standard MTT assay. The cell lines used, Caco-2 and RAW 264.7, represent gut epithelia and macrophages, which are two possible targets to encounter the NPs after oral exposure (Gastromark) or intravenous injection (Feridex, Resovist, Combidex) in NMR imaging. The analysis of the degree of survival after incubation with the nanoparticles showed that viability of cell culture was not significantly reduced by the presence of the nanoparticles up to 1 mM of iron concentration after 24 h of treatment (80–100% viability compared with the control) (Figure 7a,b). The effect of functionalization with the diamine PEG derivative was not very clear at high iron concentration. In the case of Caco-2 cells, PEG coating reduced the cytotoxicity for NP-PEG-(NH_2_)_2(2000)_ while in RAW 264.7 it had no effect.

To characterise the internalization and biodegradation of the nanoparticles, we determined by ELISA (Enzyme-Linked ImmunoSorbent Assay) the ferritin formation in Caco-2 cells as a measure of iron uptake. The NP-DMSA seems to be internalized by the cells more efficiently than PEG-coated nanoparticles visible as a 4.5-fold increase in intracellular ferritin compared with the control in contrast with a 2.7 fold increase for NP-PEG-(NH_2_)_2(2000)_ (Figure 7c). The coating with the bifunctional polyethylene glycol derivative was found to decrease nanoparticle internalization.

During the phagocytic process, macrophages generate inflammatory mediators such as chemokines, cytokines, proteolytic cascades and nitric oxide (NO) [72]. In order to evaluate the macrophages’ activation in the presence of the nanoparticles, the levels of NO were measured (Figure 7d). Endogenous NO contributes to various biological functions such as apoptosis, neurotransmission, cell vasodilation and antibacterial activity in macrophages. Lipopolysaccharide (LPS), which stimulates macrophages to produce a large amount of NO, was used as a control. A significant increase of nitrite levels released to the cell culture medium was detected for RAW 264.7 cells treated with both types of nanoparticles. Notably, PEG coating significantly reduced to a half the macrophages’ response. These results agree with other studies that evidenced that PEGylation can decrease macrophage or HeLa cell uptake of iron oxide nanoparticles where the polymer increases uptake hindrance compared with uncoated nanoparticles [39,69].

Overall, the results obtained in vitro are very interesting. The coating molecules (in this case DMSA and diamine-PEG molecules used for the functionalization) can influence the bio-interactions of the nanoparticles with the cells. In fact, even when the biomolecular interface created around the nanoparticles with the serum proteins is very similar in composition, it looks like very slight differences in the ratio of opsonins/dysopsonins can significantly affect the nanoparticles’ uptake and biotransformation. These results could explain, in part, previous observations in different animal models where NPs modified with a bifunctional polyethylene glycol derivative showed reduced absorption in the gut of *Xenopus laevis* embryos [48] and extended circulation times after intravenous injection in Wistar rats [40,41] and New Zealand white rabbits. [39] Other biological implications explained by specific binding of proteins as biomarkers is related to the different hematotoxicological pattern observed for DMSA and PEG-coated particles. In previous studies, we described a higher anticoagulant effect for NP-DMSA compared with PEG-coated ones [73]. Proteomic analysis showed that the coagulation factor X is present in the hard and soft corona of NP-DMSA and only in the soft corona of PEG-coated nanoparticles. It is possible that this stronger interaction with the surface of NP-DMSA provokes conformational changes in the protein that prevent its cleavage and activation of the coagulation cascade. These results highlight the necessity of further characterisation studies at the bio-nanointerface to ensure nanoparticle safety in biomedical application and to design more efficient nanomedicines.

## 4. Conclusions

In this work, we have investigated, by an extensive proteomic analysis, the formation and composition of the protein corona around iron oxide nanoparticles with two different coatings: dimercaptosuccinic acid (DMSA) and its surface modification with a bifunctional (PEG)-derived molecule (2000 Da). In contrast with different reports, where PEGylation reduced protein adsorption to the surface of nanoparticles, we identified a major number of proteins (136) in NP-PEG-(NH_2_)_2(2000)_ versus NP-DMSA with 121 proteins. Furthermore, we observed a relatively high homology between the composition of the soft and hard corona of the NPs. Approximately 70%, for NP-DMSA, and 65%, for NP-PEG-(NH_2_)_2(2000)_, of the proteins detected in the soft corona were also detected in the hard corona of the nanoparticles. The PEGylation of the nanoparticles did not show a significant variation in their interactions with the proteins according to their biological function. Having said that, a slight enrichment in the content of albumin in the surface of NP-PEG-(NH_2_)_2(2000)_ could be related with the results obtained in the in vitro experiments in RAW 264.7 and Caco-2 cells. The viability of both cell lines is preserved in all cases when treated with the nanoparticles up to an iron concentration of 1 mM. Above that concentration, PEGylated nanoparticles caused less toxicity on Caco-2 cells which were also more sensitive than RAW 264.7 macrophages. The NP-DMSA were absorbed more (4.5 fold increase) in comparison with NP-PEG-(NH_2_)_2(2000)_ (2.7 fold increase), suggesting that the differences observed in the uptake of the nanoparticles were related with the chemical characteristics of the coating and also with the biomolecular interface formed around the NPs with the proteins of the serum. Moreover, it was demonstrated that these two parameters were of great influence in the in vivo behaviour of these nanoparticles.

The results presented in this work indicate that the composition of the protein corona is very difficult to predict. Proteins from different functional categories (cell components, lipoproteins, complement, coagulation, immunoglobulins, enzymes and transport proteins) were identified in all samples with very small variability. At the moment, the identification of the protein corona should not be used as a substitute for relevant biocompatibility tests. But with the growing certainty that the corona composition is what is “seen” by the reticuloendothelial system, further studies are required to increase our understanding of NPs behaviour in vivo and suggest new directions in the control of the biodistribution, biodegradation and clearance of multifunctional nanomedicines.

## Figures and Tables

**Figure 1 materials-12-02218-f001:**
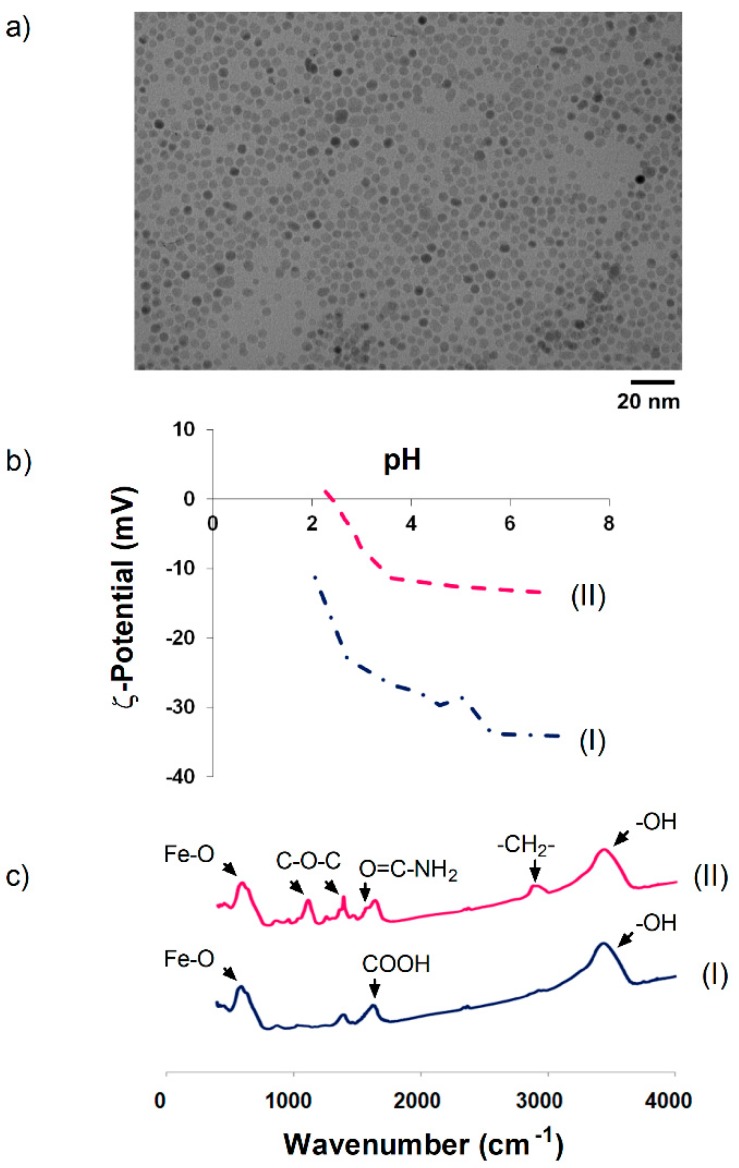
(**a**) Transmission electron microscopy (TEM) images of 10 nm nanoparticles. (**b**) Evolution of ζ-potential as a function of pH. (**c**) FTIR spectra. (I) NP-DMSA, (II) NP-PEG-(NH_2_)_2(2000)_.

**Figure 2 materials-12-02218-f002:**
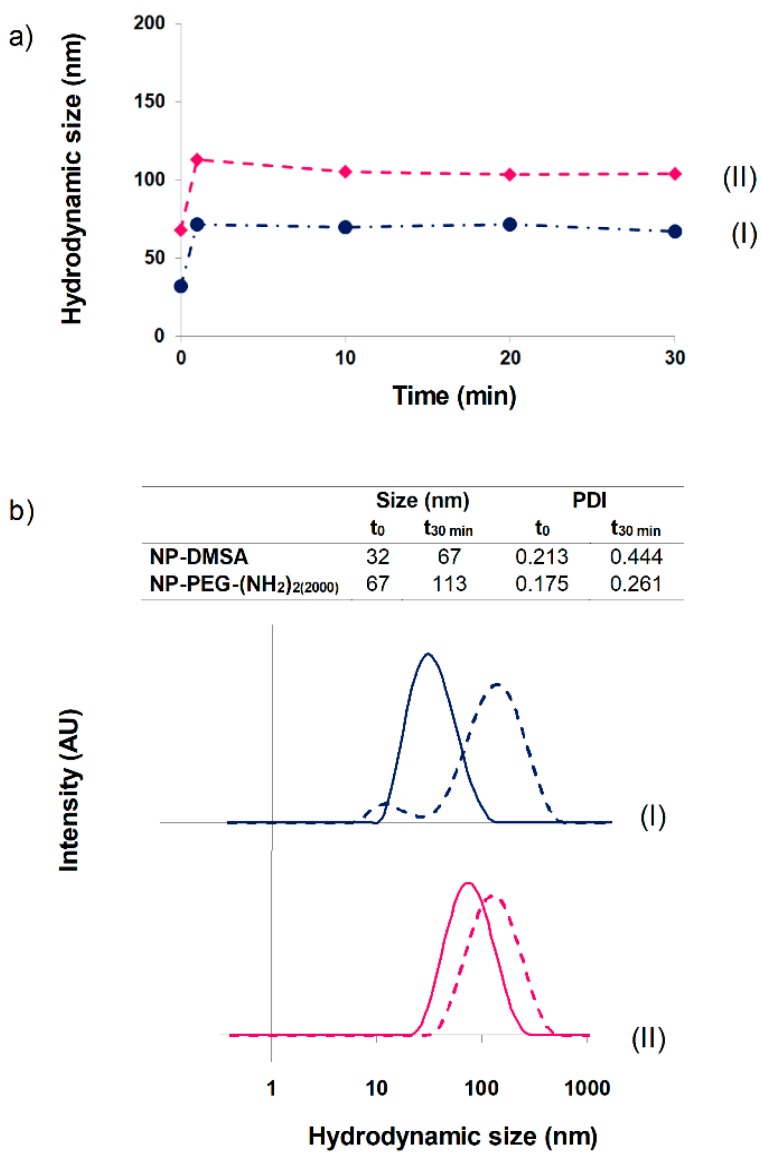
(**a**) Nanoparticle hydrodynamic size evolution up to 30 min in presence of plasma. (**b**) Hydrodynamic sizes of nanoparticles in aqueous suspension (solid line) and after 30 min of incubation with plasma (dotted line). (I) NP-DMSA, (II) NP-PEG-(NH_2_)_2(2000)_.

**Figure 3 materials-12-02218-f003:**
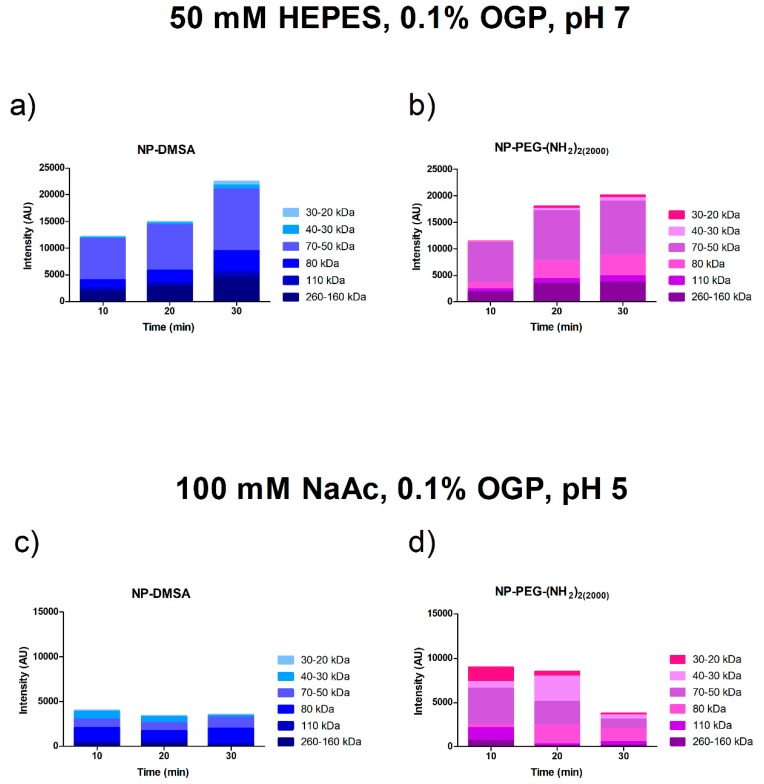
Densitometric analysis of the Coomassie blue-stained SDS-PAGE bands. Proteins from particles incubated at 10, 20 and 30 min at 37 °C were eluted sequentially with buffers at different pH. Top panel: Proteins eluted after incubation in buffer 50 mM HEPES 0.1% OGP, pH 7 (soft corona). Lower panel: Proteins eluted after incubation in buffer 100 mM NaAc 0.1% OGP, pH 5 (hard corona). (**a** and **c**) NP-DMSA (**b** and **d**) NP-PEG-(NH_2_)_2 (2000)_.

**Figure 4 materials-12-02218-f004:**
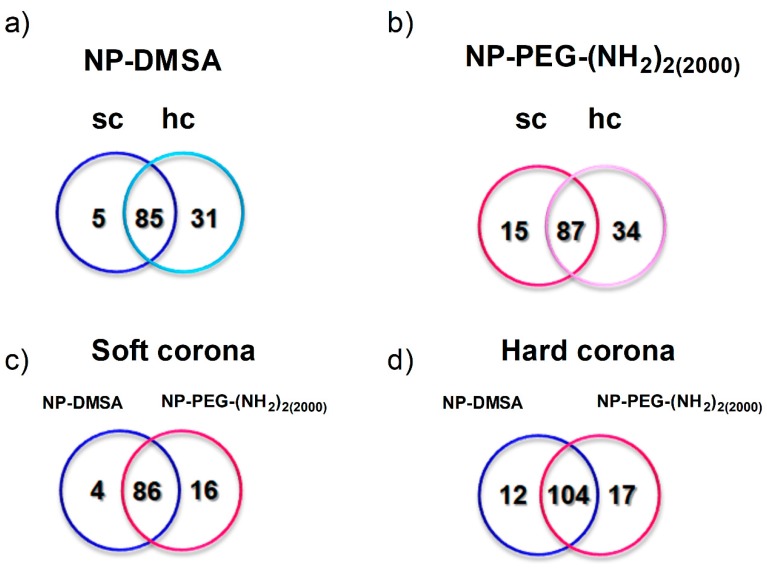
Proteins identified by mass spectrometry on the surface of the nanoparticles. Individual analysis of the protein corona around: (**a**) NP-DMSA. (**b**) NP-PEG-(NH_2_)_2(2000)_. Sc and hc represent specific proteins detected in the soft and hard corona of the particles, respectively. Overlapped areas represent proteins detected in common in their soft and hard corona. (**c**) Analysis of the proteins identified in the soft corona of both nanoparticles. (**d**) Analysis of the proteins identified in the hard corona of the nanoparticles. Blue represents specific proteins detected in NP-DMSA and pink represents specific proteins detected in NP-PEG-(NH_2_)_2(2000)_. Overlapped areas represent proteins detected in common in those fractions.

**Figure 5 materials-12-02218-f005:**
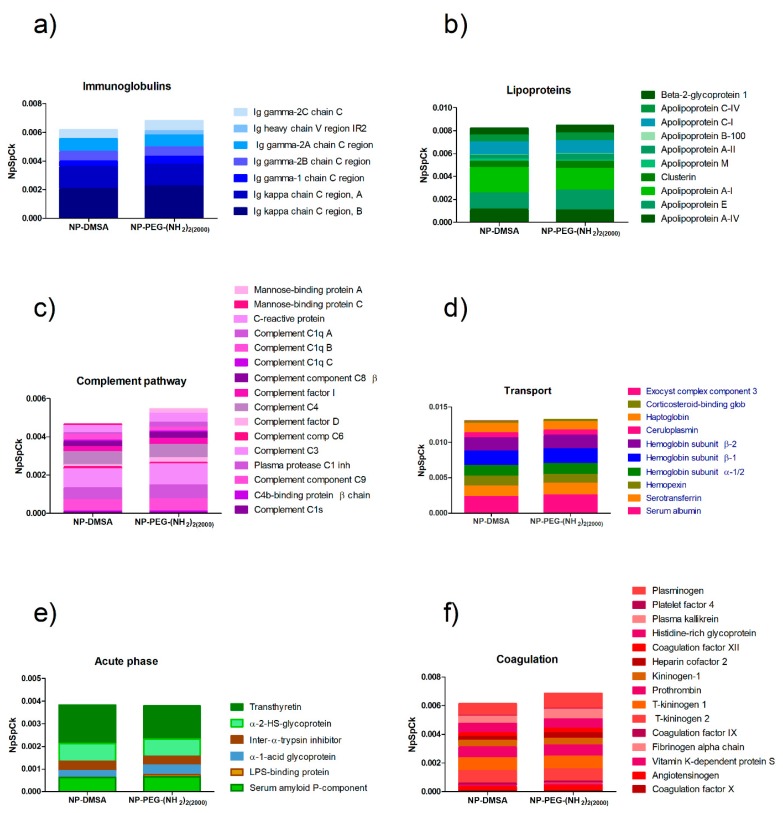
Semiquantitative analysis of the protein corona composition of NP-DMSA and NP-PEG-(NH_2_)_2(2000)_. (**a**) Inmunoglobulins; (**b**) Lipoproteins; (**c**) Complement pathway; (**d**) Transport; (**e**) Acute phase; (**f**) Coagulation.

**Figure 6 materials-12-02218-f006:**
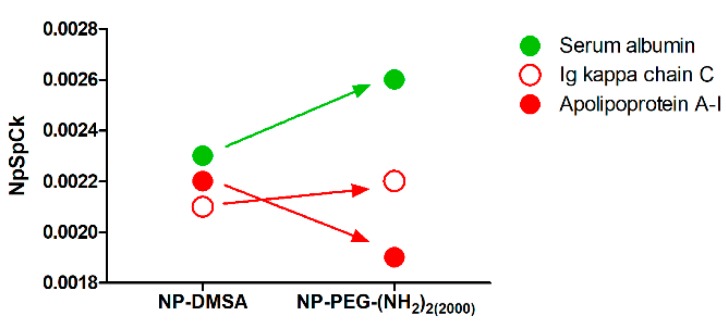
Relative abundance of opsonins (immunoglobulins and apolipoproteins) versus dysopsonins (albumin) on the protein corona composition of NP-DMSA and NP-PEG-(NH_2_)_2(2000)_.

**Figure 7 materials-12-02218-f007:**
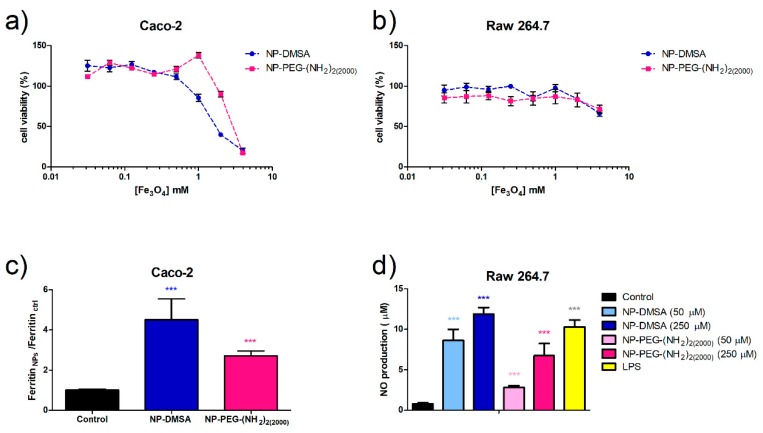
Evaluation of cell viability by MTT assay. (**a**) Caco-2 cells. (**b**) RAW 264.7 cells. (**c**) Ferritin formation measured by ELISA in Caco-2 cells exposed to 250 µM of NP-DMSA or NP-PEG-(NH_2_)_2(2000)_. (**d**) Evaluation of macrophages activation by measurement of nitric oxide production. Data represent means ± SD (*n* = 3). * Shows statistical significance compared with the control (one-way ANOVA, Bonferroni´s post-hoc test * *p* < 0.05, ** *p* < 0.01, *** *p* < 0.001, **** *p* < 0.0001).

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
