# Peer review of "Understanding the Influence of a Bifunctional Polyethylene Glycol Derivative in Protein Corona Formation around Iron Oxide Nanoparticles"

_materials, 2019, doi:10.3390/ma12142218_

Round 1
Reviewer 1 Report
The study by Ruiz et al. assesses the effect of 2 different IONPs coatings on the corona formation and the consequent nanoparticles internalization and processing by 2 different cell lines.
The methodology and results are sound and constitute yet another proof of the importance of bio-processing in understanding nanoparticles' fate.
My only remark concerns the text editing (namely the language), so I suggest the authors proofread their text (e.g. make sure that the terms absorption and adsorption are used correctly, check the use of prepositions and pronouns, punctuation, etc.).
Author Response
We thank Reviewer #1 for the comments and suggestions that helped us to improve the quality of the manuscript. In the new version of the manuscript, we have carefully revised the language and the use of the terms absorption/adsorption.
Reviewer 2 Report
The manuscript provide the influence of a bifunctional polyethylene glycol derivative in the protein corona formation around iron oxide nanoparticles.
Author Response
We thank Reviewer #2 for their evaluation of our manuscript.
Reviewer 3 Report
The manuscript entitled Understanding the influence of a bifunctional polyethylene glycol derivative in the protein corona formation around iron oxide nanoparticles by Ruiz et al. deals with proteomics of protein corona formation around iron oxide nanoparticles decorated with dimercaptosuccinic acid (DMSA) and diaminopolyethylene glycol [PEG-(NH2)2] (2000 Da). The manuscript is well structured and written with plenty experimental data.
I agree with the publication of the manuscript in the present form.
Minor aspects should be considered before publication:
1. All references indicated within the manuscript should be written correctly (e.g. [20]-[21] should became [20, 21]; [28]-[30] should became [28-30] etc).
2. There is a “Y” missed in spectrometer (line 174).
Author Response
We thank Reviewer #3 for the positive comment.
1. We have modified the reference’s style according to the guideline of the journal and now the style used corresponds to the American Chemical Society.
2. The heading on line 174 (now line 152) has been corrected.